# Image-Based Phenotyping Study of Wheat Growth and Grain Yield Dependence on Environmental Conditions and Nitrogen Usage in a Multi-Year Field Trial

**Stanley J. Miklavcic** [1], **Joshua Chopin** [1,*] **and Hamid Laga** [2]

1 Phenomics and Bioinformatics Research Centre, UniSA STEM, University of South Australia, Mawson Lakes, SA 5095, Australia; stan.miklavcic@unisa.edu.au
2 School of Information Technology, Murdoch University, Murdoch, WA 6150, Australia; h.laga@murdoch.edu.au
* Correspondence: josh.chopin@unisa.edu.au

**Abstract:** As the global population and resource scarcity simultaneously increase, the pressure on plant breeders and growers to maximise the effectiveness of their operations is immense. In this article, we explore the usefulness of image-based data collection and analysis of field experiments consisting of multiple field sites, plant varieties, and treatments. The goal of this approach is to determine whether the noninvasive acquisition and analysis of image data can be used to find relationships between the canopy traits of field experiments and environmental factors. Our results are based on data from three field trials in 2016, 2017, and 2018 in South Australia. Image data were supplemented by environmental data such as rainfall, temperature, and soil composition in order to explain differences in growth and the development of plants across field trials. We have shown that the combination of high-throughput image-based data and independently recorded environmental data can reveal valuable connections between the variables influencing wheat crop growth. Meanwhile, further studies involving more field trials under different conditions are required to test hypotheses and draw statistically significant conclusions. This work highlights some of the more responsive traits and their dependencies

**Keywords:** wheat; phenotyping; yield; nitrogen use efficiency; NUE; image analysis; crop; sustainability; nitrogen management





## 1. Introduction

One of the main goals of precision agriculture is to empower farmers with information that could allow the reduction in nutrient input costs. By eliminating excess fertiliser applications, growers can optimise their operations both economically and environmentally [1]. While one important part of sustainable agriculture is to decrease environmental degradation through minimising chemical inputs, crops still need to be of a high quality for consumption and other end-uses. From an output perspective, two major factors that cereal plant growers are interested in are the total grain yield and the grain protein content (GPC) [2]. Grain yield is influenced by a variety of phenotypic factors [3–5], including—but not limited to—plant number, plant height, canopy coverage and colour, spike number, and spike size. Meanwhile, factors influencing GPC are nitrogen (N) availability, nitrogen uptake efficiency, and nitrogen utilisation efficiency; the latter two contribute to nitrogen use efficiency (NUE) [6,7]. As nitrogen features on both sides of the optimisation equation, as a nutrient for input and as a defining factor in the quality of grain output, it is clearly one of the most important targets for resource management in the eyes of growers. Yet, while the rate of nitrogen applied to wheat globally has increased almost fivefold in the last 50 years [8–10], more than 50% is not absorbed by crops but is lost to the environment [8].

This equates to unnecessary input costs on the one hand, and increased environmental impacts on the other. Despite these alarming statistics, excessive nitrogen use remains the main nutrient input used to improve yield, although many countries have been able to increase wheat crop productivity in the absence of nitrogen and with other nutrient limitations [11,12].

A number of studies have demonstrated the variability associated with the use and uptake of nitrogen by plants in field conditions. Mahjourimajd et al. [13] reported on the effect of nitrogen across a wide ranging population of wheat tested under the field conditions of different locations; both canopy and grain yield traits were monitored in the search for the genetic and environmental factors that affect the response. The study built upon an earlier field and genetic study [14] of the effects of temperature and moisture on grain yield and grain quality. In [15], Mon et al. investigated the effect of five rates of nitrogen treatment and ten rates of irrigation on the yield and NUE of durum wheat in Arizona, USA. The results of their study suggested that utilising phenotypic traits such as canopy temperature, along with weather data, could lead to better irrigation and nitrogen management strategies. This approach to investigating the relationship between crop outcomes and the environment, as well as different resource management strategies, is not uncommon in the literature. In [16], Ravia et al. proposed that early nitrogen deficiencies had no impact on final yield and GPC, while Dumont et al. [17] showed that weather conditions between sowing and the flag leaf stage affected the optimal use of nitrogen. In [18], the spatial characteristics of the field and the SALUS modelling approach were used to optimise the use of nitrogen, resulting in increased profit and decreased impact on the environment.

The above-cited works serve to exemplify the growing awareness of and interest in understanding how the changing condition of the field during the season—as a function of environmental factors—correlates with final crop yield. Unfortunately, the time- and labour-intensive exercise of quantifying field performance as a function of time and environmental condition is far from common practice. Still, a number of attempts have been made. For example, M-Lopez et al. [2017] undertook a time-course study of a wheat trial, capturing field characteristics (vegetation indices) based on hyperspectral and thermal images taken by cameras mounted on an airplane flying 270 m above ground; while this approach is time- and labour-expedient, the level of detail that is possible to acquire may be limited with this resource (and aircrafts are not always readily available). A popular alternative [19–21] for fast crop phenotyping comes in the form of unmanned aerial vehicles (UAVs). In [22], the authors used a UAV to derive a leaf area index (LAI). The LAI at each location was then used to estimate the spatial properties of the soil which were input into a mechanistic model for providing recommendations to growers for subsequent nitrogen treatments. While UAVs are a suitable solution to some phenotyping problems, many UAVs lack the resolution necessary to solve others. In [23], the authors compare the use of a UAV and a ground-based platform for estimating crop vigour and height for different varieties and nitrogen treatments, showing that UAVs provide superior estimates of the former, while ground-based platforms perform better for the latter. Such ground-based platforms have been used frequently throughout the literature to estimate crop parameters that require a higher resolution of data, such as spike counting [24], 3D reconstruction of canopies [25], and determining growth stages [26]. More recently, the use of LIDAR as a phenotyping tool has also been tested to quantify crop traits such as canopy biomass and canopy height [27]. In the work reported below, we have chosen the more laborious but more accurate alternative of ground-based observation of wheat fields in order to capture as much detail as possible.

In summary, in this paper, we direct attention to the challenge of quantifying the relationship(s) between environmental factors, such as soil and weather, and yield. Moreover, we employ a high-throughput, image-based approach to in-season field phenotyping to explore the time-dependent relationships between developing crop characteristics, such as crop canopy coverage and crop vigour, and yield. In the process, we identify the role of environmental factors influencing these relationships. Using small representative fields

of 60 and 90 plots, we demonstrate the potential to draw insights from a rich dataset of multi-year, multi-site, multi-treatment, and multi-variety trials. Our approach uses a manually propelled, land-based vehicle equipped with several synchronised cameras. Data from the images are analysed with a variety of image analysis algorithms explained herein. These image-based results are then further analysed and discussed in relation to the environmental data to tell a more complete story about crop yield, grain GPC, and plant NUE over the course of three years and three field sites.

## 2. Materials and Methods

This experiment was conducted over three subsequent years, 2016, 2017, and 2018, at three different—though geographically adjacent—field sites in Mallala, South Australia. In this section, we provide a description of the field trials for each year, the image acquisition process, and the data collection details.

**Field trials.** Plants of 10 Australian varieties of spring wheat (*Triticum Aestivum* L.) were planted in single plots of $4 \times 1.2$ m$^2$. The ten varieties used were the following: Drysdale, Excalibur, Gladius, Gregory, Kukri, Mace, Magenta, RAC875, Scout, and Spitfire. In 2017, the variety Yitpi was used in place of Spitfire due to issues with availability. The first trial, in 2016, contained 60 plots, while the second and third trials, in 2017 and 2018, respectively, contained 90 plots.

In 2016, the 60 plots (located at latitude = $-34.45°$, longitude = $138.48°$) were laid in a 12 row by 5 column configuration, containing six replicates of each variety, three of which were treated with fertiliser and three left untreated. The trial was sowed on July 8 and the 37.5 g/m$^2$ : 4.3 g/m$^2$ of $16 - 8 - 16 \, N - P205 - 1 < 20$ nitrogen fertiliser, as well as urea, were applied on the 12th of August 2016. The ten varieties were distributed in a split-plot design, with a semi-random, semi-systematic arrangement. No weed or insect control treatment was applied during any of the growth seasons.

In 2017 (located at latitude = $-34.47$, longitude = $138.48$) and 2018 (located at latitude = $-34.44$, longitude = $138.47$), the 90 plots were laid in a 18 row by 5 column configuration, containing 9 replicates of each variety. Three replicates were treated with fertiliser at an early developmental stage, three were treated at a later developmental stage, and the remaining three were left untreated. The same fertiliser and urea concentrations and amounts were applied in 2017 and 2018 as those which were applied in 2016. Moreover, the same split-plot design was used, just with six more rows. In 2017, the trial was sowed on July 3, an early treatment of fertiliser was applied (to 30 plots) on July 14, and a late treatment was applied (to 30 other plots) on September 26. In 2018, the trial was sowed on June 1, early fertiliser treatment was applied to 30 plots on June 21, and a late treatment was applied to 30 other plots on August 21. It is notable that sowing in 2018 occurred approximately one month later in the year than sowing in the respective years of 2016 and 2017. The task of choosing an optimal sowing time depended on a number of factors, such as weather, climate conditions and forecasts, soil moisture, and rainfall patterns. As such, the domain experts responsible for sowing the trials use their best judgement, combined with the data at hand, to decide on the sowing date each year.

**Image acquisition.** A hand-manoeuvred imaging platform, composed of a robust steel frame on a dual-axle undercarriage with inflatable wheels, was propelled through the field once weekly, or as often as weather would permit. A stereo pair of Canon EOS 60D digital cameras (manufactured by Canon, Shimomaruko, Ōta, Tokyo, Japan; purchased in Adelaide Australia) was mounted centrally on the frame, approximately 20 cm apart and two metres above ground level. In contrast to the first two cameras, whose optical paths were angled vertically for horizontal stereo viewing of plots, a third and fourth camera were mounted at the two extremities of the platform for the purpose of ensuring the oblique viewing of the plots. A schematic of the platform is shown in Figure 1.

Manual focus was used during all imaging sessions with cameras focused at 2 m and 1.5 m during the early and late plant growth stages, respectively. The camera settings were as follows: focal length—18 mm; aperture—f/9.0; ISO—automatic and exposure time—

1/500 s. Cameras were synchronised to capture images within 1 millisecond of each other. Three images were taken of each plot, at the beginning, centre, and end of the plot, so that coverage, vigour, and height data could be averaged over these images for consistent measurements. The details of how these images are used to collect data about plant canopy coverage, vigour, and height are provided in the next section.

**Soil, yield, and quality data.** Soil analyses were conducted prior to sowing each field trial. Soil samples were consistently taken from a depth of 0–10 cm and analysed after being dried to 40 °C. The soil analyses returned 45 outputs, including a range of different variables such as the amounts of specific elements, the soil colour, and the composition of soil types.

Crop yield and quality data were gathered post-harvest in 2016 and 2018. Unfortunately, in 2017, all plots were heavily affected by a disease that prohibited the acquisition of meaningful yield and grain quality data. Yield and quality data, including grams per plot, grain moisture, and grain protein content for 2016 and 2018, were recorded for each individual plot, allowing for an average over the three replicates of each variety–treatment combination. Henceforth, this average will be used when discussing the results, not only for yield and grain quality, but also for the image-based data of coverage and height.

**Weather data.** All weather-related data were obtained from the Australian Bureau of Meteorology (BOM) website [28]. The nearest BOM weather station to our field trials is located at Roseworthy, South Australia (latitude = −34.53°, longitude = 138.75°). While data for daily rainfall were complete over the time span of our field trials, some data points for the minimum and maximum temperature were missing. On these rare occasions, the nearest BOM weather station with data for those days was used. The nearest BOM weather station is located at Edinburgh, South Australia (latitude = −34.71°, longitude = 138.62°). In each of these cases, the surrounding temperature recordings from Edinburgh and Roseworthy were compared to ensure that there were no vast differences in the measurements.

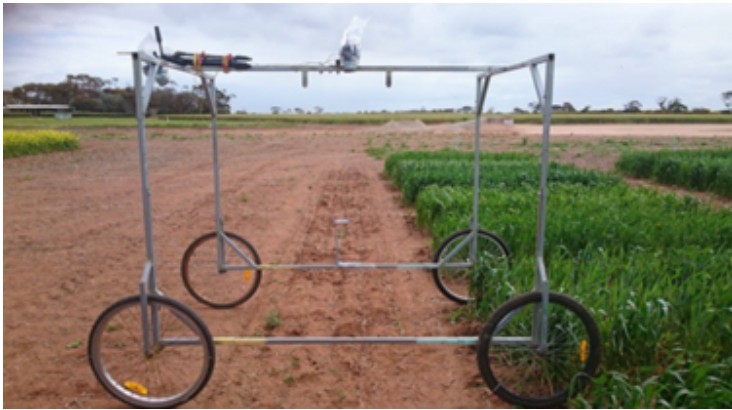

**Figure 1.** The platform used to capture images of plants in the field. The manually propelled platform is comprised of a custom-built steel frame and four bicycle tyres. A colour chart for calibration and up to five cameras can be mounted on it (stereo top view, stereo oblique view, and one side view).

**Crop height data.** Height was recorded manually with the same technique by the same individual at regular times over the course of each of the three seasons. A one-metre ruler with centimetre markings was sequentially placed at uniform points along each plot to determine sample heights and, consequently, average plot height. Sample heights were defined as the average height of the top of the spikes in line with the ruler at each sample position; if no spikes were present, then the tallest leaves were used.

*Image Processing*

As mentioned in the previous section, two of the three groups of data used in our analysis—canopy coverage (as a percentage of area) and canopy vigour—were derived from the images. The third set of data, that of canopy height, was recorded manually

(although this could also be estimated from the stereo images, as was the case in [29]). Here, we give a brief overview of the methods used to obtain these data. For full details on the image processing pipeline for estimating coverage and height, the reader is referred to [29,30], respectively.

**Crop coverage estimation.** Our approach for coverage estimation involves three steps: segmenting green plant pixels, cropping a region of interest (ROI), and calculating the proportion of the ROI that is covered by green plant pixels. To segment the plant pixels—hereafter referred to as the foreground—from the image background (soil, vehicle parts, etc.), we used a support vector machine (SVM), a supervised machine learning technique which attempts to find the best hyperplane that separates a dataset with two classes. The training data for SVM are a labelled set of pixel values, belonging to either the foreground or the background class, enabling the method to find the best hyperplane or boundary that will provide the maximum classification accuracy for future candidate pixels. Rather than segmenting high-resolution images manually to obtain the training data or manually—and subjectively—selecting small subsets of pixels from those images, we used K-means clustering to select them semi-automatically. Using K-means clustering, each training image is segmented into 20 clusters with minimal intra-class variance. Each cluster is then manually given a label as green plant or background. The centre of each cluster, or mean colour, is then used as the training data for the SVM. A visual representation of the trained SVM can be seen in Figure 2.

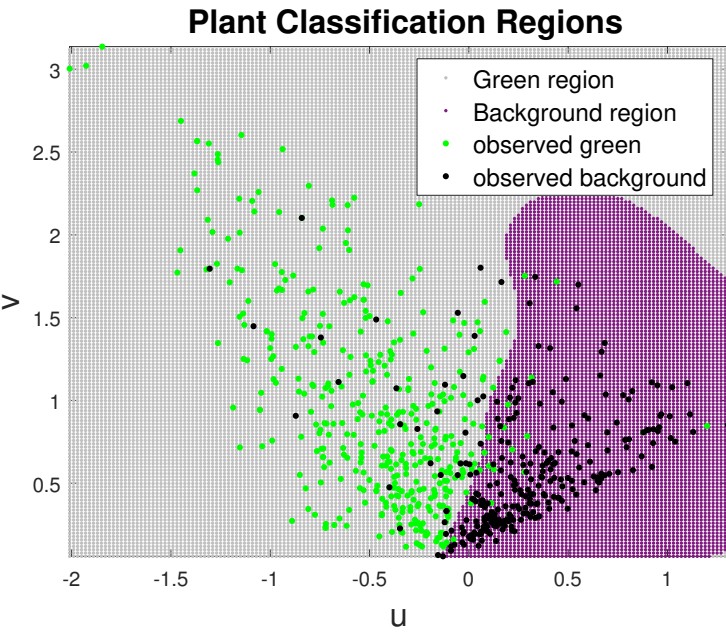

**Figure 2.** Illustration of support vector machine (SVM) trained on plant images. The *x* and *y* axes refer to the *u* and *v* channels of the Luv colour space, respectively. The large green and black scatter points refer to observed pixels of green plant material and background, respectively, used for training. The SVM algorithm "learns" the boundary between grey and magenta regions.

The region of the image that will be analysed, hereafter referred to as the region of interest (ROI), was chosen to be a rectangular region with a height that is the full height of the image and with a width that is defined as being parallel to and slightly inside the vehicle's rails—see Figure 3b for a visual illustration. The vehicle's rails were detected with a combination of greyscale thresholding and hough transforms for finding straight lines in the image post-thresholding. On occasion the algorithm for automatic rail detection produced erroneous results, which were easily and automatically detected by comparison with correct detection. In these instances, the ROI was manually determined by assigning the four coordinate pairs of the ROI rectangle. An example of an original and segmented

image, with the ROIs highlighted, can be seen in Figure 3. The final remaining step is to divide the number of pixels that were determined to be green plant material (i.e., plant pixels) within the ROI by the total number of pixels within the ROI.

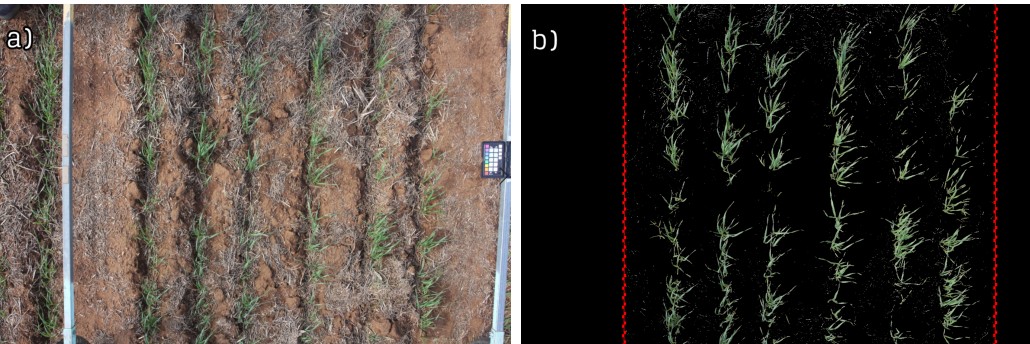

**Figure 3.** Original and segmented image. (**a**) Image of a single wheat plot during an early stage of the 2017 field trial. Within this image, there are regions, such as the platform's rails, the colour chart, and a neighbouring plot, that should not be used in determining this plot's coverage. (**b**) The same plot after segmenting out only the plant pixels. The dotted red lines are approximately parallel to the platform's rails and used to determine plate coverage as a proportion of the total region of interest (ROI) area.

**Vigour estimation.** To estimate plant canopy vigour, we use the green plant pixels segmented from the background in the *coverage estimation* section. The red and green channel values of each plant pixel contribute to a plot average. Denoting the average red and green plant pixel values by $R$ and $G$, respectively, an estimate of vigour, $V$, can be given by

$$V = \frac{G - R}{G + R}. \tag{1}$$

The vegetation index in Equation (1) is commonly referred to as the green–red vegetation index (GRVI) and has been chosen to estimate vigour for a number of reasons, such as light intensity normalisation and proven correlations with important phenotypic traits. A more in-depth survey of different vegetation indices and the strengths of GRVI can be found in [23].

## 3. Results and Discussion

### 3.1. Field Conditions

The first piece of the puzzle in understanding the relationships within our multi-faceted dataset is daily rainfall. Rainfall is perhaps the most important environmental factor when considering plant growth in the field. Given the number of drought-tolerant varieties used in these trials, rainfall will be an important factor in understanding plant growth and development across varieties. Figure 4a–c show the daily rainfall from May 1 (always before sowing) to November 30 (always after harvest) for the years 2016–2018, respectively. Note in particular that, in both 2016 and 2017, a significant amount of rain fell shortly before (2016) or just after sowing (2017): 35 mm over three days prior to sowing in 2016, and 23 mm 16 days after sowing in 2017. In 2018, however, a significant amount of rain fell only 24 days before sowing and 64 days after sowing. The latter amounts were 12.8 mm and 16.2 mm, respectively. Figure 4d shows the cumulative rainfall over the same yearly interval in each of the three years. The 2016 and 2017 seasons received large volumes of rainfall shortly after sowing. This is shown in the cumulative graph.

We draw particular attention to the significantly different pattern of the 2016 field trial compared with either of the 2017 or 2018 trials. The 2016 trial is distinguished by a largely continual rain pattern resulting in a near-linear cumulative rainfall. In direct contrast, the 2017 and 2018 trials experienced long dry periods of little or no rain at various critical

periods during the season; the longest period in both seasons lasted 60 days or more at the end of the season. It has already been mentioned that the 2018 trial also experienced a long dry period at the start of the season, after sowing and after the early fertiliser application.

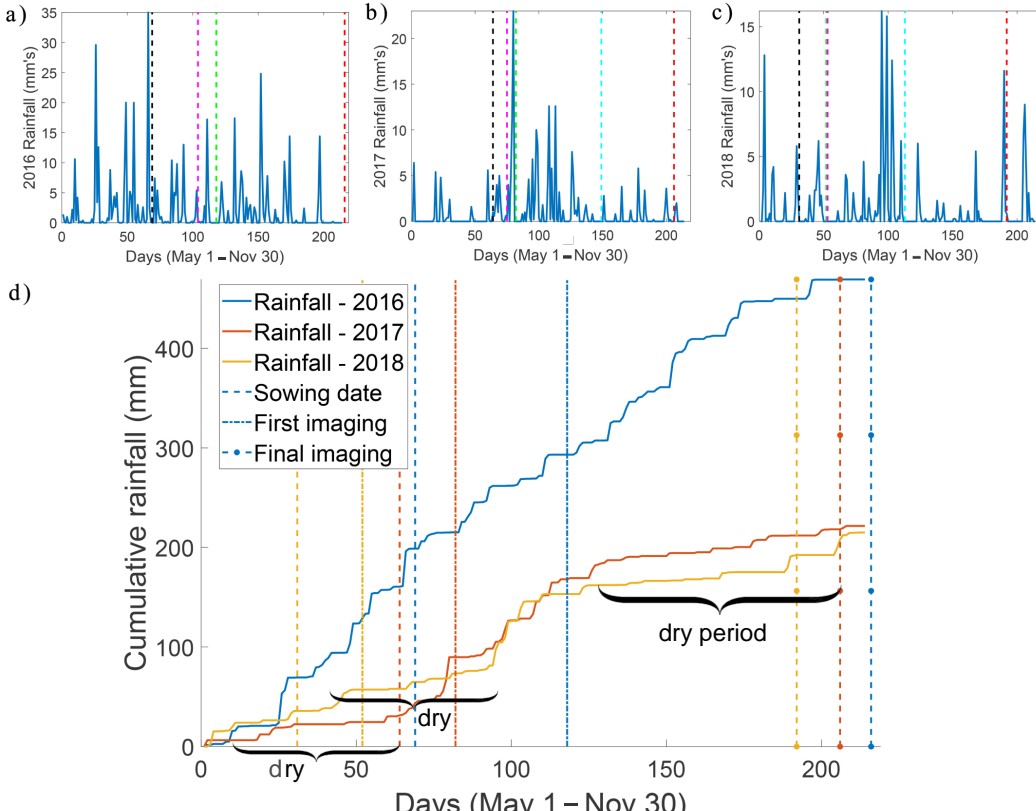

**Figure 4.** Rainfall data for the three field trial years. Panels (**a**–**c**) show the daily rainfall in mm at the 2016, 2017, and 2018 sites, respectively, for the period extending from just before sowing until harvest. The solid blue curve shows the amount of rainfall for each day, the vertical dashed black, magenta, blue, green, and red lines depict the dates of sowing, early and late fertilisation events, and the first and last days of imaging, respectively. Panel (**d**) shows the cumulative rainfall over the common period containing the three field trials. The underbraces highlight substantial dry periods during the 2017 and 2018 seasons.

To complement the rainfall data, we have also gathered data on maximum temperature recorded at the sites for each day of the same period. Figure 5 shows a moving mean of daily maximum temperatures from May 1 to November 30, for all trials. The moving mean is defined as the arithmetic mean of the temperatures of the day and the temperature of the preceding nine days. From the figure, a number of points emerge. For example, in 2017 the average temperature reached over 30 °C near the end of the season. In both 2017 and 2018 the crops regularly experienced significant periods of high temperature days over the course of the season than in 2016. Furthermore, in 2017, there was a short burst of warm temperature days shortly after sowing (approx. 60 days), which was not experienced during the 2016 and 2018 trials.

Using these two sets of weather data, we may already begin painting a picture of the environmental conditions experienced by the three trials. In 2016, conditions were more favourable, including a steady volume of rain and fewer hot days overall. In both 2017 and 2018, conditions were less favourable, but comparable; rainfall was less frequent and of lower volume and the temperatures were higher overall. A possibly critical difference between these two seasons was the absence of rainfall around the time of sowing in 2018.

The different conditions, on the one hand, and the similarities, on the other, may aid in our understanding of the phenotypic data presented later in this section.

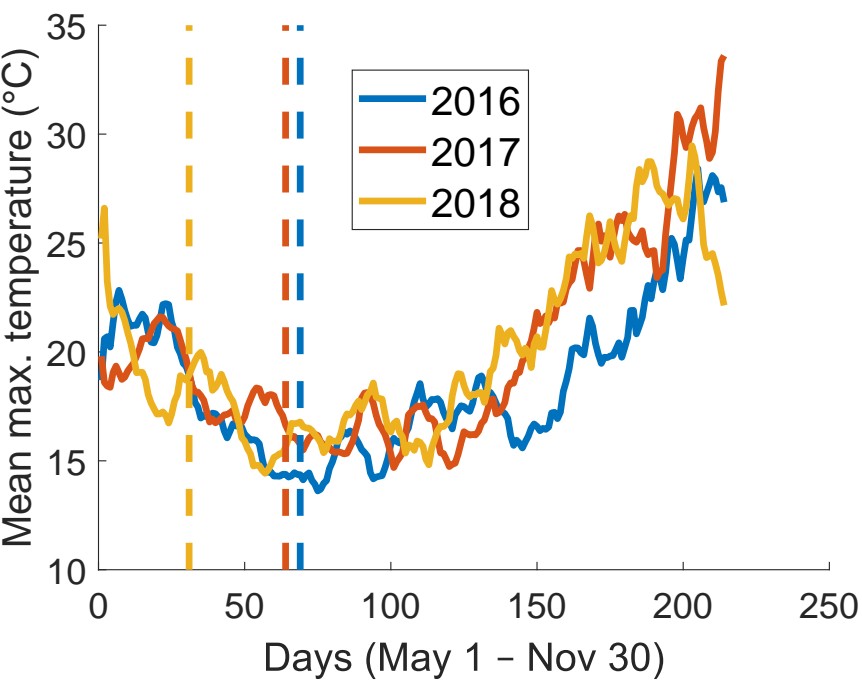

**Figure 5.** Temperature data over all field trial years. The blue, red, and yellow lines show a moving mean of the maximum temperature in degrees Celsius at the 2016, 2017, and 2018 sites, respectively, for a period from just before sowing to harvest. Each data point is calculated using the mean of the maximum temperature on that day and the previous 9 days, making a 10-day moving window. Vertical dashed lines indicate sowing dates.

Figure 6 summarises the soil characteristics for the three sites. The six (6) quantities that exhibited the greatest variation across the three sites or between at least two sites are shown. Despite their geographical proximity, differences in soil character can still be significant. In most—but not all—aspects, the soils at the 2017 and 2018 field trials were more alike. Notable exceptions are the soil composition (both silt and clay levels were higher by a factor of 2 and 1.5, respectively, at the 2018 field site) and higher level of soil sodium at the 2018 field site (a factor of 2 higher than at either the 2017 or 2016 site). These exceptions may have some bearing on the observed differences in the outcomes of the 2017 and 2018 field trials. In fact, as shall shortly be revealed, the yield outcome of the 2018 trial differed significantly from both of the two earlier trials, the latter two of which were commensurate. This comparison suggests that soil differences may nevertheless have a non-negligible, albeit second- or higher-order, influence after rainfall and temperature.

The convolution of adverse soil characteristics and low rainfall (in particular) modifies the effectiveness of water uptake by plant roots. Sandy soils of low clay content (such as the soil in the 2016 trial) are more porous and less restrictive of rainfall water movement through the soil to the roots. In contrast, the transport of water through the soil in the 2018 trial, characterised by its higher clay content and hence its lower porosity, is more restrictive. The clay composites present in the 2018 soil swell readily during periods of rain and retain their hydrated state for longer during dry periods (a so-called higher field moisture capacity [31]). In other words, the permanent wilting point is higher in the high-clay-content case of the 2018 trial compared with the 2016 trial. Thus, not only is rainfall reduced in 2018 compared with 2016, but the water availability for uptake is reduced compared with the condition of the 2016 trial. It is also worth pointing out that the higher concentration of mobile $Na^+$ ions (contributing to the higher soil electrical conductivity in the 2018 trial) may also introduce a detrimental salt stress element, given the propensity of plant roots to take up $Na^+$ in the transpiration stream [32]—wheat is known to be a salt-sensitive plant [33]. A salt stress component would only compound any drought stress component.

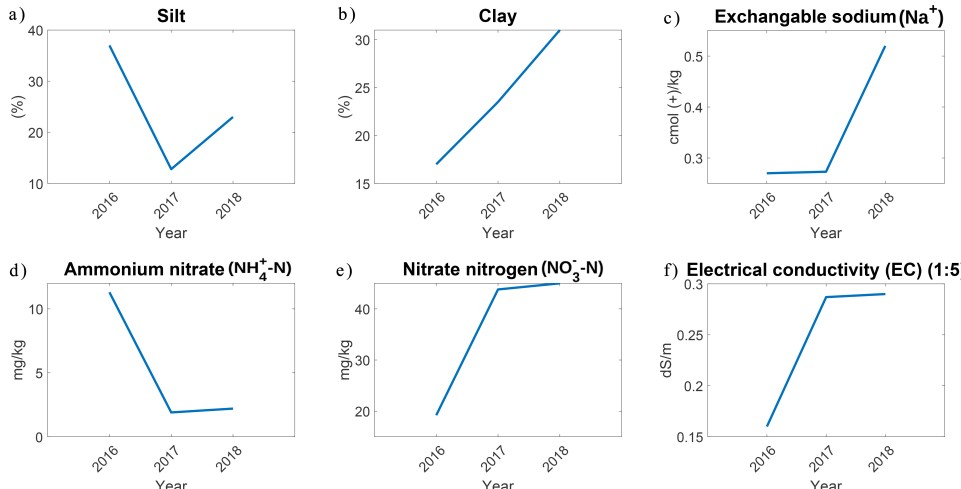

**Figure 6.** Results of soil analysis at each field trial. (**a–f**) The amount of of silt, clay, exchangeable sodium, ammonium nitrate, nitrate nitrogen, and electrical conductivity, respectively, at each of the three field trials.

### 3.2. Canopy Growth and Development

To illustrate the great distinction between the outcomes of the three years, in Figure 7, we show images taken of an arbitrary plot from each year's trial. In panels (a–c), the images were each taken (from the same height and with the same camera settings) at 90 DAS in the 2016, 2017, and 2018 field trials, respectively. Panels (d–f) show images of the same plots on the final day of imaging in those trials. These images help to understand the difference found in the quantitative data presented in all remaining figures.

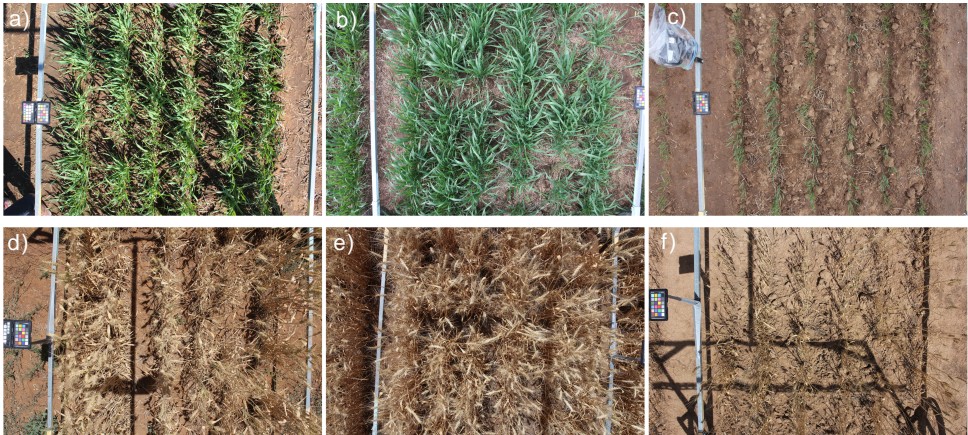

**Figure 7.** Illustrative images of random plot canopies in the respective field trials. The images in panels (**a–c**) were taken at the same time point, 90 DAS, in the respective years of 2016, 2017, and 2018. Panels (**d–f**) show images of these same plots on the last day of imaging in the respective seasons.

Figure 8 shows the canopy coverage of each plot over the three seasons. Here, the coverage is expressed as the number of green (plant) pixels in the ROI divided by the total number of pixels in the ROI. For each combination of variety and treatment, the curve presented is the average over the three replicate plots. To maintain clarity, error bars have not been included in the main parts of panels (a–c). An example of the variation across replicates is shown in the inset to panel (c). The first immediately obvious point of difference between the three summary figures is that the canopy coverage was least in 2018 by a substantial margin. At their most dense, the plot canopies in the 2016 and 2017 trials almost cover the entire ROI (near 100% coverage). In contrast, the greatest coverage achieved in the 2018 trial was approximately 25%. The distinctly different canopy coverage

situations in 2017 and 2018 are particularly interesting as the cumulative rainfall patterns and temperature data appear very similar for the two years. One possible explanation for the different canopy performance between these two years may be the amount of rain that fell specifically around the respective times of sowing, as previously discussed in connection with Figure 4. Reflecting on the points made earlier in this section (see *Field conditions*), the higher clay content, and the mobile $Na^+$ concentration in the 2018 trial, compared with the conditions of the 2017 trial, may also go some way in explaining the difference in the overall outcomes of these trials, which otherwise had relatively similar rainfall and temperature patterns. Explicitly, it is quite possible that the combination of poorer water penetration, greater clay swelling, increased soil water retention (higher permanent wilting point [31]), and increased mobile $Na^+$ in the plant-available water could have also contributed to the poorer outcome in 2018.

We draw attention to a distinct and sharp drop in coverage appearing in the 2017 trial around 100 days after sowing (DAS). Recall that the values shown are based on green (plant) pixel count. Consequently, this drop in coverage means that, at around 100 DAS, the canopy began a sudden and significant increase in senescence, irrespective of variety and treatment. This behaviour was exhibited by all the repeats. The sudden change in leaf condition is distinctly different from the more gradual and variety-specific transition to a senescent state exhibited by the 2016 trial. The sudden onset of senescence in the 2017 trial follows a significant dry period, starting at around 65 DAS, which continued until the end of the imaging period (see the right-most underbrace in Figure 4). This behaviour is expected because, in general, increased senescence is known to occur when a dry period takes place after the anthesis stage [34].

Although plotted on the same scale (Figure 8a,b), the 2017 trial differs from the 2016 trial in not showing any great separation between varieties or treatments. In fact, not only did the untreated plots perform as well as the treated plots in 2017, they performed significantly better than the untreated plots in the 2016 trial. A possible reason for this might be found in the degree of preexisting soil (nitrate) nitrogen in 2017 (which was more than twice the soil concentration found in the 2016 trial). The lack of rain in 2017 would then not be an impediment to the availability of nutrients to the plots (in 2017).

In Figure 8, we retained the same specific scale for the sake of comparison across seasons. This choice, however, made it difficult to discern differences in performance between varieties in the 2018 trial due to the latter's low coverage values. In Figure 9, we show a magnified version of the canopy coverage specifically for the 2018 plots. Here, we see a distinction between fertilised and unfertilised plots. Plots that were treated early achieved a relatively larger canopy coverage than the plots that were treated later in the season, which in turn performed better than the untreated plots. The result indicates that, despite the overall low (green pixel) coverage in this season, the plots responded more strongly to early fertilisation. The difference in outcomes, incidentally, shows that—in a field setting—early fertilisation is much more advantageous than late fertilisation, at least as far as the canopy is concerned; this is because there is a greater chance of surface-applied fertilisation being taken up at some point in the season.

In Figure 10, the average height of plots across all varieties in all three seasons is shown. As with the response of canopy coverage, we find, as expected, the early-fertilised plots to be the tallest in all three seasons. However, the difference is the greatest in the 2016 season. In 2017, the difference in height between treatments was minimal, again in line with the canopy coverage finding. In 2018, the difference was greatest at the first recorded point and from there converged to zero by the end of the season. In this last season, the first height measurement was recorded a large number of days after sowing, compared with the previous two seasons. This was due to the slow growth early in the 2018 season. Since plant height was measured manually, discerning differences in the heights of plants shorter than 20–25 cm proved challenging. As a result of this practical limitation, this lower limit became the minimum height recorded for all height measurements. In the 2018 trial, the slow rate of plant growth meant that the canopy height reached this threshold value

much later in the season (>100 DAS), and only notably increased above this level following a substantial period of rainfall (between 100 and 120 DAS, shown in Figure 4).

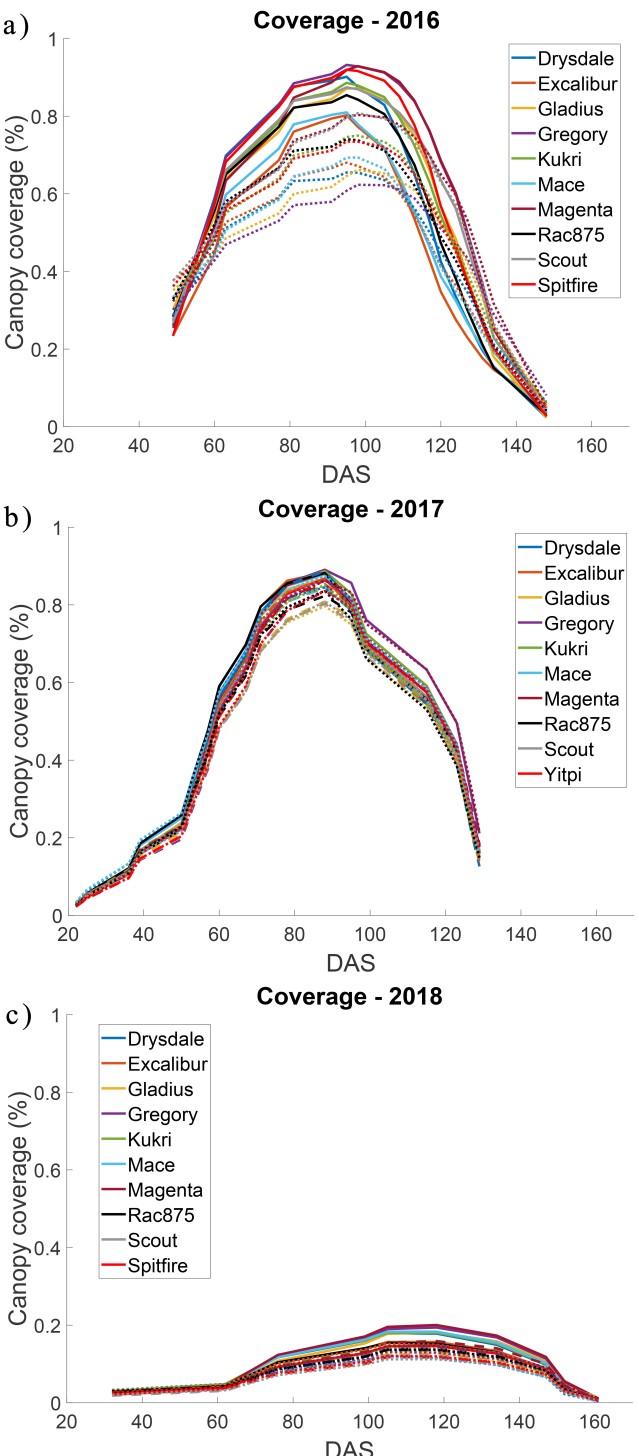

**Figure 8.** Plots of canopy coverage expressed as a percentage of the total region of interest (ROI) that is occupied by green (plant) pixels. Panels (**a**–**c**) show canopy coverage for the 2016, 2017, and 2018 field trials, respectively. Varieties are identified by colour, as indicated in the legend. The different curve styles refer to different treatments: solid curves refer to plots treated early in the season, dashed curves refer to plots treated later in the season, and dotted curves refer to untreated plots. In 2016, there was no late treatment of plots.

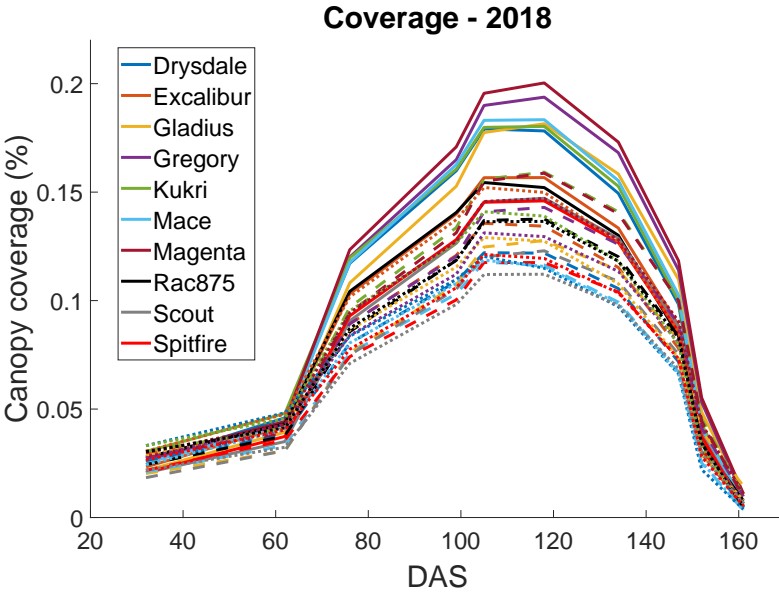

**Figure 9.** Magnified version of canopy coverage for the 2018 field trial. Varieties are identified by colour as given in the legend. The different curve styles refer to different treatments: early treatment (solid curves), late treatment (dashed curves), no treatment (dotted curves).

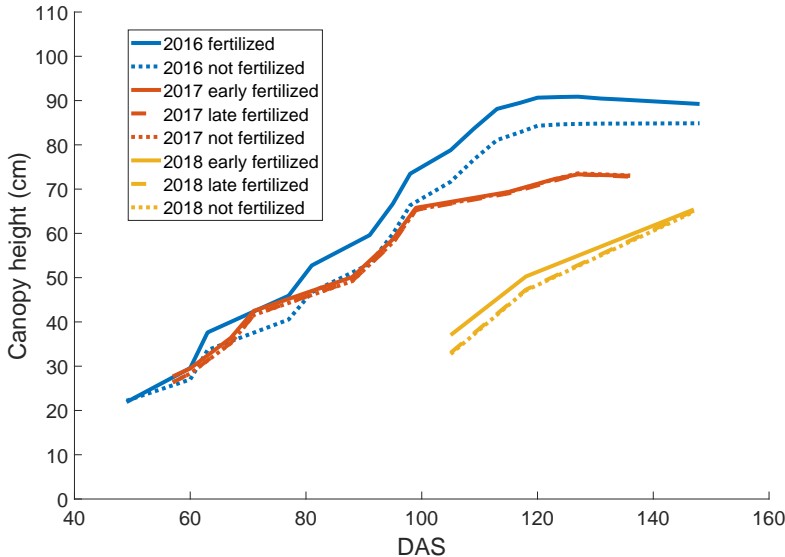

**Figure 10.** Average canopy height in cm. The individual curves show the average over all varieties and repeats for a given treatment condition. Solid lines represent plots that were fertilised early in the season, dot–dashed lines represent plots that were fertilised later in the season, and dashed lines represent unfertilised plots. The colours refer to the different years: (2016) blue, (2017) red, and (2018) yellow.

Figures 8–10 provide an overall perspective of the difference between fertilised and unfertilised plots across the three trials. However, discerning differences between individual varieties requires a more focused study. Given that there are ten varieties (Yitpi in place of Spitfire in 2017)—and therefore $3 \times \frac{10 \times 10}{2} = 150$ distinct comparisons to make—for convenience, we have singled out two varieties for explicit review, that displayed markedly different responses to fertiliser treatment. The focus is again on canopy coverage and canopy height. Information about other specific variety pairs can be found in the Supplementary Data File S1.

Figure 11 shows canopy coverage and canopy height as a function of DAS for the varieties Drysdale and Scout. Growth indices such as canopy height and coverage are mea-

sures of clorophyll content [35] and hence can be used to assess not only plant growth but also the nitrogen status of the canopy. As was true of all varieties, both coverage and height were significantly affected in an adverse manner in the 2018 season when compared with the previous two seasons. However, we may formulate additional hypotheses concerning the NUE of these two varieties over the three seasons.

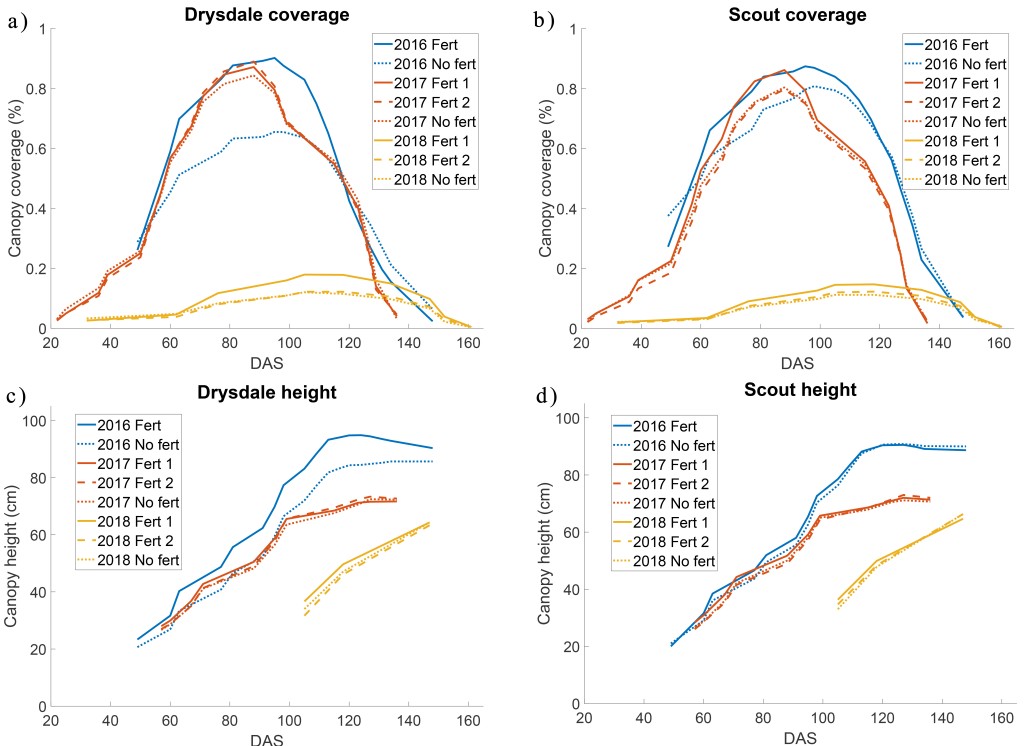

**Figure 11.** Canopy coverage and height, respectively, as a function of time and fertiliser treatment for two varieties: Drysdale (panels (**a**,**c**)) and Scout (panels (**b**,**d**)). Solid lines refer to averages over replicate plots with early treatment of fertiliser; dashed lines refer to plots with late treatment of fertiliser; and double-dashed lines refer to unfertilised plots. Colour coding again refers to the different trial years: 2016 (blue), 2017 (red), and 2018 (yellow).

In 2016, the differences in canopy coverage between fertilised and unfertilised plots of both Drysdale and Scout were significant, peaking at 28% and 18%, respectively—approximately at the booting developmental stage of each. With regard to canopy height, while there was a discernible difference in the average heights of fertilised and non-fertilised Drysdale plots in 2016, treatment did not appear to affect the average canopy height of the Scout variety. Given that this analysis is derived from data taken during the same season, at the same location, and hence under the same soil conditions, it would appear that Drysdale is either more effective at taking up nitrogen or better at utilising the nitrogen that it does take up to develop both canopy and height; meanwhile, Scout is either less effective in taking up nitrogen or it only relocates nitrogen for leaf production (or both).

In 2017 and 2018, the differences in canopy coverage and height across treatments for the two varieties were less pronounced but generally consistent with the picture adopted by all varieties. To be more specific, the 2018 trial showed the same relative trend as that appearing in 2016, for both varieties, except at a greatly reduced magnitude. In comparison, the response in the 2017 trial is somewhat the converse of the 2016 trial in that Scout appeared to have the larger response to nitrogen, although still weaker than in 2016. On the other hand, in 2017, Drysdale did not improve its performance with treatment in either aspect of canopy height or coverage. It is again to be noted that the untreated plots of both varieties performed better in the 2017 trial than the corresponding untreated plots of the 2016 trial. As already hypothesised, this could be attributed to the higher concentration of

nitrate nitrogen found naturally in the soil at the 2017 site, as shown in Figure 6e. Drysdale is known for its high water use efficiency and is an excellent variety when selecting for drought tolerance [36]. This may explain its superior ability in a year (2018) with little rainfall around the time of sowing, improving coverage with fertilisation, compared with the improvement found with Scout.

We shall deduce more information about the growth of these two varieties, as well as the remaining eight, using data from the yielded grains.

To supplement the canopy coverage and canopy height data, we have also derived estimates of vigour from the images. Figure 12 shows the GRVI measure of vigour for all like-treated plots from the 2017 and 2018 field trials. Unfortunately, issues during image acquisition in 2016 resulted in substantial inconsistencies in illumination, which could not be corrected due to the absence of a colour reference chart [30]. Hence, vigour estimation using individual colour channels was not possible. Recall that the GRVI values shown in Figure 12 are derived from colour channel values for plant pixels specifically, and then averaged over the plot prior to averaging over all replicates and all varieties. It is therefore not possible to make statements on GRVI differences between varieties. Moreover, the graphs do not quantify canopy coverage, only quality aspects of plant pixel colour indicating its state of health. Low absolute values of GRVI are likely to reflect near-equal contributions from green and red channels; high positive values reflect a predominance of green over red, while the converse is true for large negative values. We point out that, while plant segmentation is certainly accurate enough for both canopy coverage and vigour estimation in the middle and late stages of plot growth, inaccuracies may occur at an early stage. Small weeds and dead plants in the background are sometimes segmented as 'green plant pixels', and while these errors make up a tiny percentage of the total pixel values in the middle and late stages of data capture, they may have a larger relative effect on coverage and vigour values at very early stages. This challenge will be further addressed later in the 'future directions' section.

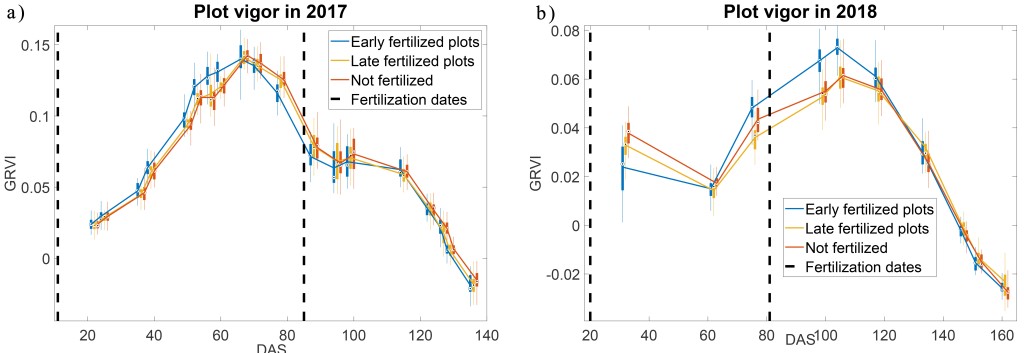

**Figure 12.** Estimated average vigour of plots as a function of time based on the green–red vegetation index (Equation (1)). The data have been averaged over all varieties and replicates undergoing the same treatment. For convenience, we highlight the two extreme years, 2017 (panel (**a**)) and 2018 (panel (**b**)). Vertical dashed black lines show the dates at which fertilisation occurred (early fertilisation line appears close to the vertical axis); blue and yellow curves show the vigour of fertilised plots (early and late, respectively); red curves show the vigour of unfertilised plots. The average GRVI for each wheat variety is illustrated using a box plot showing the median and the 25th and 75th quartiles. The whiskers of the box plot extend to the most extreme data points, excluding outliers.

For the most part, the trends in vigour shown by both sets of graphs follow those displayed by the canopy coverage graphs. Vigour is low but positive on average in the early stages, likely indicating equal green and red channel values, and increases to the peak approximately in the middle of the plant's life cycle before decreasing again toward harvest time as the degree of senescence increases. Since it is known that wheat plants exhibit early

vigour [37], we may attribute the low positive values seen in early stages of plant life in these plots to artefacts due to the above-mentioned inaccuracies in image data.

Overall, within each of the 2017 and 2018 seasons, the plots seemed to reach peak vigour at approximately the same time point, regardless of when fertiliser is applied, or if it is not applied. However, on closer inspection, it is the case that—even on average—the plots that received early fertiliser treatment did generally have somewhat higher vigour values, and reached peak vigour somewhat earlier than those treated later or not at all. In fact, the later-treated plots have vigour profiles that are all but indistinguishable from the untreated plots. This is consistent with our earlier findings on the effect on canopy coverage and canopy height of a late treatment of fertiliser.

Comparing the outcomes across the seasons, we see considerable difference in the timing of events as well as the vigour profiles. The crops in the 2017 and 2018 seasons reached their respective peak vigour values at significantly different times: around 65 and 105 DAS, respectively. While there is a ten-day difference between the first applications of fertiliser in 2017 and 2018, it is unlikely that this would account for the 40 day difference, in terms of DAS, between the peak vigour events of those seasons. It is more likely, again, that the difference is due to the different weather and soil conditions in those seasons. In terms of rainfall, the aforementioned lack of water around the time of sowing in 2018 could result in the plants' delay in reaching peak vigour. Furthermore, in 2017, a significant period and amount of rain (approximately 23 mm of rain three days after the first fertiliser treatment) may have facilitated the quick uptake of fertiliser and thus accelerated the vigour and growth of those plots generally. The differences in soil conditions, interacting with rainfall, could also have contributed to the delay in reaching peak vigour. We may again refer to the increased soil fraction of clay (in particular) in the 2018 trial compared with the 2017 trial. Although the increase is only approximately 50% (from 20% of overall content to 30%), this may have been sufficient to increase the field water capacity as well as the permanent wilting point of the soil, leaving less plant-available water. The latter could also have been made available considerably later due to the slower release of absorbed water by the clayey soil. On this note, Lal and Shukla [31] cite an initial rapid absorption of rain water due to clay aggregate swelling, followed by a slower but sustained period of absorption. During a subsequent dry spell, the dehydration of a wet clay would then be correspondingly slow. Both factors may have contributed to the delay in nutrient uptake by plant roots, since uptake predominantly occurs in the presence of water flow.

One of the more notable differences between the vigour profiles of the two seasons shown in Figure 12a,b is the appearance of a plateau in the 2017 season, beginning around 80 DAS, which is absent in the 2018 trial. This plateau translates to a knee-bend in the canopy coverage profiles (present also in the plant height curves) and consistent with a change in leaf condition resulting from a lengthy dry period beginning around 65 DAS. It is likely that this interrupted the normal plant growth behaviour and initiated an early onset of senescence. Another point of differentiation, also consistent with the canopy coverage results in Figure 8, is the scale of the vigour graphs. The 2018 profile overall shows significantly lower vigour values compared with the 2017 trial (approximately 50%). This is consistent again with the behaviour of the other phenotypic traits captured in this season.

While it is of course possible to compare plant vigour profiles between varieties, this quantity shows greater fluctuation on a variety level than does canopy coverage, which makes a definitive comparison more difficult. We have not pursued a detailed comparison here.

As stated in the Introduction, optimising crop growth involves a mixture of minimising nutrient input (costs) and maximising crop output. As such, the relationships between crop treatment, grain yield, protein content, and moisture content lie at the heart of the optimisation problem. In Figure 13, we present post-harvest data from the 2016 and 2018 seasons for some selected varieties. Unfortunately, all plots of the 2017 field trial were affected by disease at the end of the season. Consequently, yield data for 2017 were

unavailable. Still, many valuable insights can be garnered by considering these graphs, supplemented by the environmental and image-derived data presented so far.

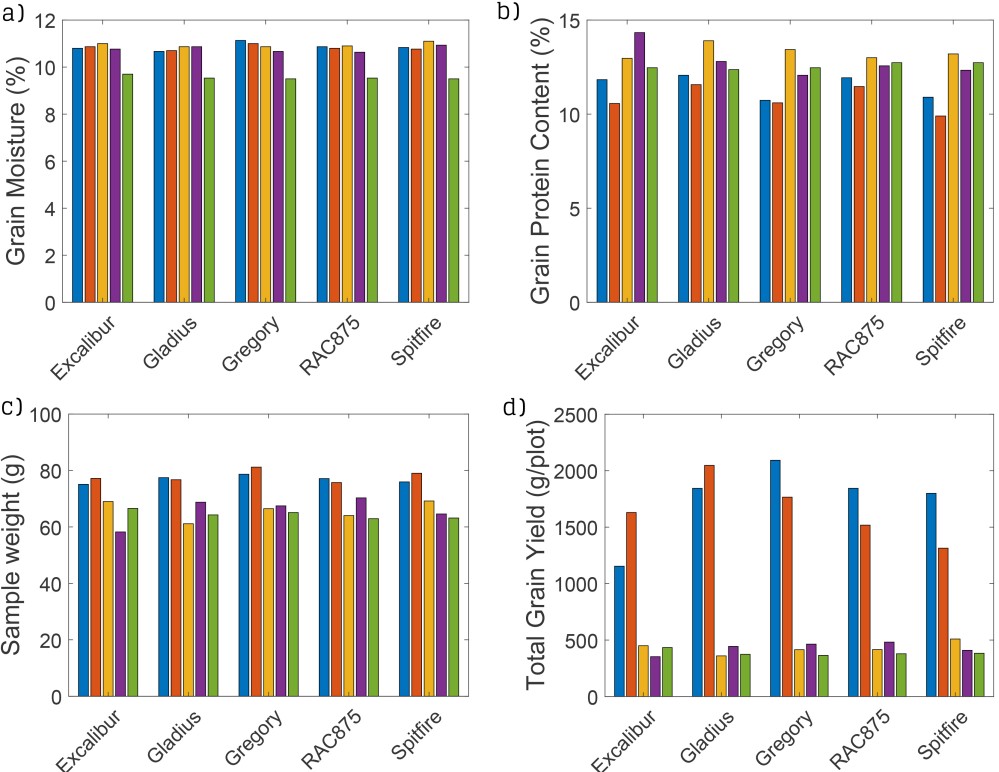

**Figure 13.** Grain yield and quality data. The four plots show values for (**a**) moisture content, (**b**) protein content, (**c**) test weight, and (**d**) total weight, separated by treatment and year. The bars are colour-coded as follows: blue is 2016 fertilised plots, red is 2016 unfertilised plots, yellow is 2018 early-fertilised plots, purple is 2018 late-fertilised plots, and green is 2018 unfertilised plots.

First—not surprisingly—the total weight of the grains, i.e., the plot yield, from the 2018 trial was much lower than that from the 2016 trial. Again, this is consistent with a poor field outcome demonstrated by our data on coverage, height, and vigour. Looking at the yield in 2016, we can see that each variety reacted differently to fertilisation. Excalibur and Gladius actually performed better without fertilisation, while Gregory, RAC875, and Spitfire experienced significant improvements in yield when fertilised.

The moisture content of grain often affects its quality and sale weight; according to a US standard (DuPont Pioneer [38]), wheat should always be harvested between 14% and 20% moisture. Figure 13 shows that, on average, all varieties in 2016 and 2018 were under the recommended levels of grain moisture. We note that the moisture contents of all unfertilised varieties were considerably lower in 2018. In contrast, the 2016 data shows no discernable obvious difference between the fertilised and unfertilised plots. This is consistent with all the results presented so far, where plants in the 2018 season were starved for water around the time of sowing and relied more heavily on fertiliser. At the level of individual varieties, Excalibur is known to exhibit a fast recovery from a period of drought stress. This is confirmed in Figure 13, where it is seen to be among the best-performing varieties in terms of moisture content, when fertilised in the water-deficient year of 2018.

While all varieties performed poorly in 2018 in terms of moisture content and total yield, in terms of grain protein content, every variety fared better: the GPC measure was generally higher compared with the 2016 trial. There is consensus in the literature that yield and grain protein content exhibit an inverse relationship [39]. This makes the task of genetic screening for a variety that is both high-yielding and whose grains are high in protein challenging [40]. From Figure 13b, the most consistently performing varieties

in terms of GPC is RAC875. As with all varieties, the yield performance of RAC875 was significantly reduced in 2018. However, the GPC was the highest of all varieties in 2016. The consistency in GPC of RAC875 could be attributed to the varieties noted to have an exceptionally low transpiration rate [41]. To see complete plots containing the data of all varieties, the reader is directed to Supplementary Data File S1.

### 3.3. Statistical Analysis

In Section 1, we alluded to the ambition of not only investigating the relationship between environmental conditions and crop outcomes, but also quantifying the relationship between in-season-dependent variables and crop outputs. In this section, we pursue those aims. The wide range of input variables (years, varieties, treatments, and weather conditions) and output variables (coverage, height, vigour, grain yield) allow for the possibility of collecting different subsets of the data to address different relationship questions. We explore a few of these possibilities here.

A partial general picture of the effect of environment, unfiltered by wheat variety, is shown in Figure 14. Here, we show eight time series of correlation coefficients, four summarising the time-dependent correlation between canopy coverage (green pixel count/plot area) and grain weight (gm/plot or tonnes/hectare) (Panel (a)) and four summarising the time-dependent correlation between canopy vigour (GRVI) and grain weight (Panel (b)). The curves differentiate between years and state of fertilisation, but are averaged over varieties and replicates of varieties.

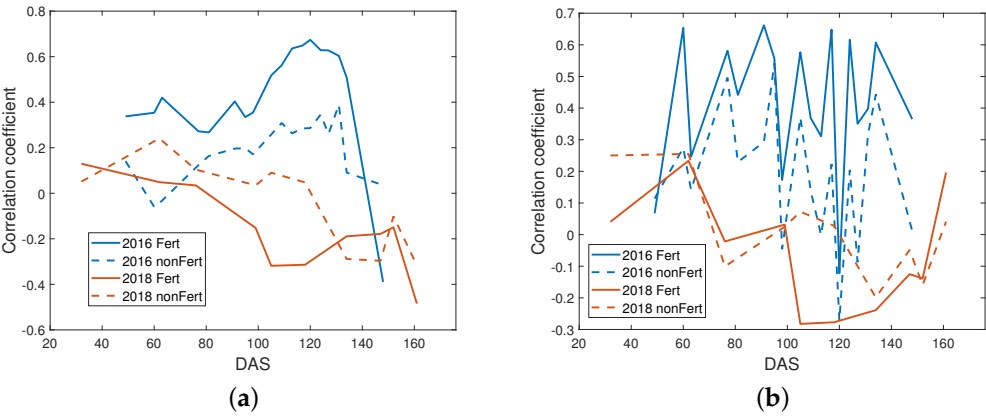

**Figure 14.** Measures (correlation coefficients) of the correlation between canopy coverage and recovered grain weight (panel (**a**)) and the correlation between canopy vigour and recovered grain weight (panel (**b**)) for the 2016 (blue curves) and 2018 (red curves) seasons (no grain was recovered in 2017 due to crop infection). Solid curves refer to (early-) fertilised fields; broken curves refer to unfertilised fields.

While the well-watered field condition in 2016 stands out as the most important factor in terms of coverage, showing a significant positive correlation index, even more prominent is the inevitable consequence of the well-watered condition: the strong response to the application of fertiliser (in 2016). An equally important point conveyed by the figure is the converse: that, in 2018—which suffered from a lack of adequate rainfall throughout the season—there is an absence of a correlation, if not a negative correlation, between coverage and yield. The evidence for rainfall being a key factor is supported by the negative correlation values of vigour versus yield in 2018. Although the outcome of the inconsistent illumination conditions in 2016 is plain from the two erratic time series shown in Figure 14b, there is nevertheless a consistent trend in the two 2016 series to support the claim of the overarching positive influence of rainfall, encouraging the uptake of fertiliser.

What may also be deduced from Figure 14 (at least, panel (a)) is the appearance of a crucial period in the 2016 season. In the mid–late period of the season, beginning around 100 DAS, the significant positive correlation between coverage and yield points to the

practical importance of ensuring an adequate and continual supply of water around this time. On the other hand, under less favourable conditions, as in the dry season of 2018, the notable negative correlations (obvious in both panels (a) and (b)) might indicate that plants will tend to sacrifice canopy development and health in order to favour grain production, however low that may be.

Again, by co-opting all varieties and replicates into a single-year quantity, we may examine the impact of an early application of fertiliser as a function of time more specifically, and compare and contrast the results across the three years of the study to see how other external factors affect things. A convenient and succinct measure with which to compare the distributions (as a function of time) is the Cohen's $d$ value, which is a measure of effect that places emphasis on the difference between the mean values of the two comparable distributions (those with and without fertiliser), suitably scaled by the weighted average of their standard deviations. That is, the Cohen's $d$ value is defined by

$$d = \frac{\text{difference of means}}{\text{pooled standard deviation}} = \frac{\overline{x}_1 - \overline{x}_2}{s}, \tag{2}$$

where,

$$s = \sqrt{\frac{(n_1 - 1)s_1^2 + (n_2 - 1)s_2^2}{(n_1 + n_2 - 2)}}, \tag{3}$$

and where the individual standard deviations are defined as

$$s_i^2 = \frac{1}{n_i - 1} \sum_{j=1}^{n_i} \left(x_{i,j} - \overline{x}_i\right)^2, \quad i = 1, 2. \tag{4}$$

Cohen's $d$ values for the distributions that have been averaged over variety and replicates of varieties (error bars show the extent of the variation) are calculated at corresponding time points (for each season) to give the time-dependent quantities shown in Figures 15 and 16.

Panels (a) and (b) show, in chronological detail, Cohen's $d$ values of canopy coverage (Figure 15) and canopy vigour (Figure 16) for the years 2016 and 2018, respectively. Panel (c) shows summary heat-map charts for all three years, in which the detailed time series data (exemplified in panels (a) and (b)) have been collected in coarser-grained sample bins for comparative convenience.

Although Cohen's $d$ values may be positive or negative, depending on the relative magnitudes of the means, we specify the category of their effect size based on their absolute value. In panels (a) and (b) (as well as for the corresponding case of 2017), we have adopted the convention of classifying (and colour coding) the influence of fertiliser into the following categories: no effect ($|d| < 0.2$); small effect ($0.2 < |d| < 0.5$); medium effect ($0.5 < |d| < 0.8$); and large effect ($|d| > 0.8$). Considering these categories of effect size, along with the positive and negative values depicted, a total of seven possible categories are created.

Consistent with our earlier observations, we quantitatively see that, in all three years, an early application of fertiliser has the largest influence on canopy coverage, which is manifested midway through the season. Furthermore, consistent with earlier statements, in comparing panels (a) and (b) of Figure 15 with panels (a–c) of Figure 4, we see the cooperative influence of rainfall. In the case of the 2016 data, we note that the periods when the effect of fertiliser is classed as medium–large coincide with the periods when there was not only a significant amount of rain, but a sustained period of significant rainfall. These sentiments are supported by the corresponding quantitative measures for vigour in Figure 16. As quantified by the Cohen's $d$ value, a late application of fertiliser had either no effect or only a small effect on coverage in the 2017 and 2018 seasons.

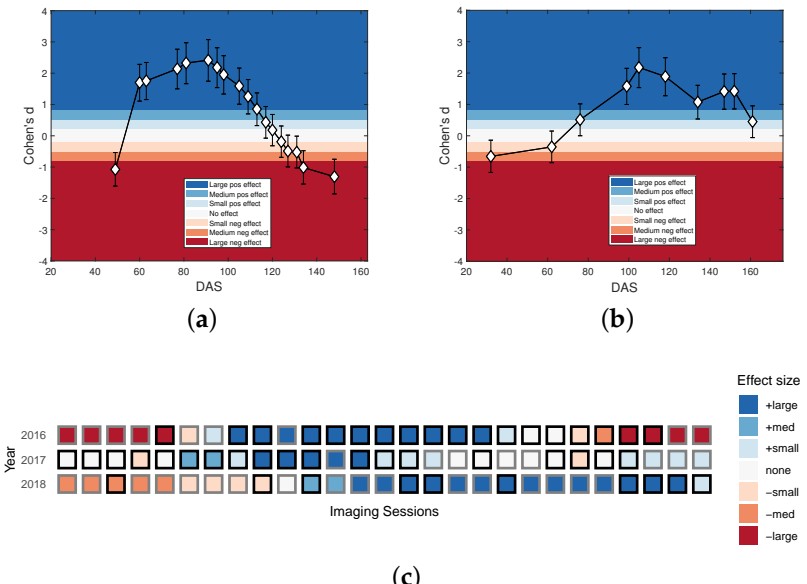

**Figure 15.** Quantitative measure of the effect of an early application of fertiliser on canopy coverage, as provided by the Cohen's *d* index of Equation (2). Panel (**a**) shows the detailed time series of Cohen's *d* values for the 2016 season. Panel (**b**) shows the corresponding detailed time series for the 2018 season. Panel (**c**) shows a coarse-grained summary heat map of all three seasons, based on the detailed data shown in panels (**a**,**b**) as well as the corresponding time course result for 2017. Squares indicating blocks of time into which explicit data fall are shown with black borders. Squares with light grey borders allude to discrete time blocks continuing a trend, but in which no data were available. The same colour coding applies across all three panels, as documented in the legends.

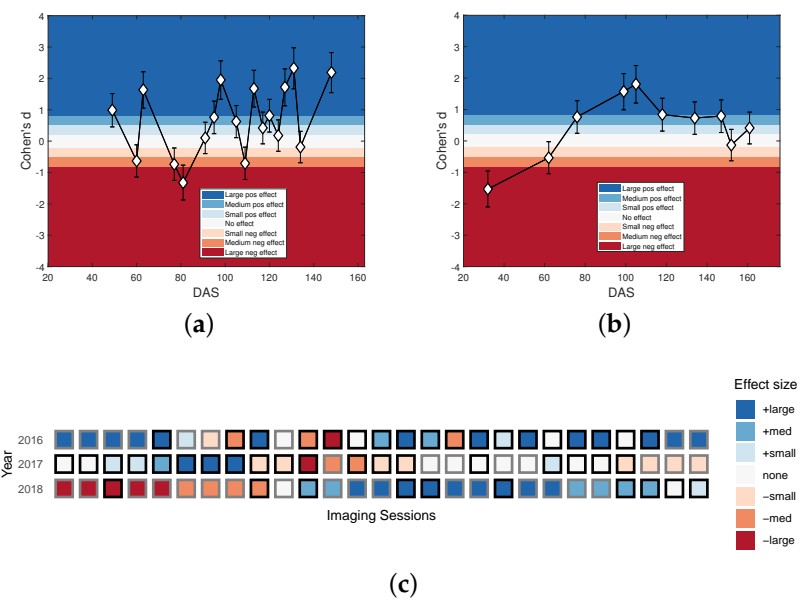

**Figure 16.** Quantitative measure of the effect of an early application of fertiliser on canopy vigour, as provided by the Cohen's *d* index, Equation (2). Panel (**a**) shows the detailed time series of Cohen's *d* values for the 2016 season. Panel (**b**) shows the corresponding detailed time series for the 2018 season. Panel (**c**) shows a heat-map summary of Cohen's *d* values for all three years; blocks are coarse-grained time collection bins of data points. Squares indicating blocks of time into which explicit data fall are shown with black borders. Squares with light grey borders allude to discrete time blocks continuing a trend, but in which no data were available. The same colour coding applies across all three panels, as documented in the legends.

## 4. Conclusions

In the ideal scenario of an experiment played out multiple times under a set of well-prescribed conditions, any measurement would, within acceptable limits, result in the same outcome. Such could be expected of glasshouse experiments conducted under controlled conditions. Our study, on the other hand, highlights the difficulty that is inherent to field experiments. Field studies are subject to fickle environmental conditions that may differ from site-to-site, season-to-season, and may often change within a single season. Drawing unequivocal conclusions on the performance of a given variety subject to a given treatment is challenging, and indeed may not even be possible. For example, we have found that, under favourable conditions—as were present in our 2016 Australian winter wheat trial—it is possible to discern performance differences between varieties and, on a more detailed level, differences in response to different fertiliser treatments for a given variety. Yet, in less favourable circumstances—e.g., under the low-rainfall–high-temperature conditions of our 2018 trial—such differences are, explicably, more difficult to discern. We have found that the distinction (between the 2016, 2017, and 2018 trials) is not simply a matter of degree. As a case in point, in 2016, the difference in canopy coverage between the fertilised and unfertilised plots of Drysdale was considerable, while for Scout it was not (Figure 11). However, in 2017, fertiliser application had a relatively small effect for Drysdale, while for Scout it was relatively large. Such variability in response advocates the need to run multi-year and multi-site trials in order to fully comprehend the influences of all contributing factors. That is, it is simply not sufficient to draw any conclusion about relative performance of any variety (or collection of varieties) from a single field trial. On the other hand, given the magnitudes and directions of the differences we saw in our multi-year trials, we would also conclude that it would be a mistake to simply average results from these multiple trials. The resulting loss of correlative detail with environment would, as a consequence, diminish the predictive power of any model of varietal performance.

To sum up, we are able to draw the following general conclusions: The well-known premise that fertiliser application leads to better yield needs qualification. Speaking only in terms of topical fertiliser application, we found that extrinsic conditions strongly influence the fertiliser uptake efficiency. In the absence of sustained rainfall (or irrigation), fertiliser application is likely to be ineffective (and wasteful) overall. Along the same lines, we found that an early application of fertiliser is preferred to a late application. It remains to be seen whether an early application simply allows for sufficient time for the necessary quantity of rain to fall in order for effective fertiliser uptake, or whether it represents a coordinated timing with a critical stage of plant development. To reiterate: complicating factors aside, our results would suggest overall that an early fertiliser treatment is preferable to a late treatment, and better than no treatment. This applies to canopy performance (coverage, vigour, and height), to grain yield, and to grain quality (GPC). However, on the subject of grain quality, there may be some varietal preference for a late treatment, as in the case of the variety Excalibur (Figure 13). However, further trials are needed to confirm this isolated outcome.

Our findings also bring to the fore the second-order influence of soil condition as a further discerning factor. The preexistence of soil nitrogen has a beneficial effect, while soil salinity has an equally obvious detrimental influence.

In our analysis, we have attempted to assess the performance of wheat crops in close connection with environmental conditions. Our efforts, in terms of producing correlation measures and effect measures, are incomplete; nevertheless, they underscore the need for detailed in-season monitoring of crop growth and development, as well as the need for a longitudinal and quantitative characterisation of environmental conditions. This firstly enables an understanding of the outcomes of a trial and secondly contributes to a database on which a predictive model of yield under any given climatic condition can be based.

Although the combination of multiple trait data gathered in this study has proved useful in an analysis of the growth and development of wheat in plant breeding trials, the range of information as well as the quality of this information could be further enhanced.

Firstly, image analysis techniques for processing a wider range of phenotypic traits exist. For instance, the number of spikes (and potentially the size of spikes) per unit area, the time of development stages, and the onset and rate of increase in senescence per plot could all be estimated based on noninvasive imaging and subsequent image analysis. In particular, improving the accuracy of segmentation at all growth stages would help reduce the relatively higher errors found during the early stages, and simultaneously allow for senescence quantification in later stages. Secondly, increasing the number of environmental variables being monitored, including more weather- and soil-related variables than were considered here, may prove to be important. A more extensive and rigorous correlation analysis between field crop performance and a wider range of environmental variables than the extent presented here could offer greater insight into the plant–environment relationship. Finally, the field trials used in this article were of a relatively small size when compared to actual plant breeding trials. Larger sample sizes of plant varieties, treatments, and environments could provide for statistically significant results about the correlations between variables.

**Supplementary Materials:** The following are available online at https://www.mdpi.com/article/10.3390/su16093728/s1, Supplementary Data File S1.

**Author Contributions:** Conceptualization, S.J.M. and H.L.; methodology, S.J.M., J.C. and H.L.; software, J.C.; validation, J.C. and S.J.M.; investigation, S.J.M., J.C. and H.L.; formal analysis, J.C.; resources, S.J.M.; data curation, J.C.; writing—original draft preparation, J.C. and S.J.M.; writing—review and editing, S.J.M. and J.C.; visualization, J.C.; supervision, S.J.M.; project administration, S.J.M.; funding acquisition, S.J.M. and H.L. All authors have read and agreed to the published version of the manuscript.

**Funding:** This research was funded by research grants (LP140100347 and LP150100055) from the Australian Research council under its Linkage grant scheme.

**Institutional Review Board Statement:** Not applicable.

**Informed Consent Statement:** Not applicable.

**Data Availability Statement:** Dataset available on request from the corresponding author.

**Acknowledgments:** The authors acknowledge help from Jinhai Cai, Pankaj Kumar, and Stephan Haefele in the acquisition of the images at various times during the 2016–2018 seasons.

**Conflicts of Interest:** The authors declare no conflicts of interest.

### Abbreviations

The following abbreviations are used in this manuscript:

| | |
|---|---|
| NUE | nitrogen use efficiency |
| ROI | linear dichroism |
| UAV | unmanned aerial vehicle |
| GRVI | green–red vegetation index |
| SVM | support vector machine |
| N | nitrogen |
| GPC | grain protein content |
| LAI | leaf area index |
| BOM | Bureau of Meteorology |
| DAS | days after sowing |

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
