# Peer review of "Image-Based Phenotyping Study of Wheat Growth and Grain Yield Dependence on Environmental Conditions and Nitrogen Usage in a Multi-Year Field Trial"

_sustainability, doi:10.3390/su16093728_

Round 1
Reviewer 1 Report
Comments and Suggestions for Authors
This paper is well organized. Good introduction and literature review. Adequate methodology and well exposed result. Extensive bibliography.
I recommend the following revision:
1.Please explain why is the sowing date in 2018 one month earlier than in previous years?
2. Line 190. In this part I suggest to include discussion of the effect of precipitation and temperature on soil properties.
3. I suggest concluding the work also with implications and limitations of the research.
Best regard
Author Response
Thank you for your valuable comments, below we will respond to each point.
1. Thank you for highlighting this important difference in the sowing dates between trials. We have added the following sentences to the materials and methods section
“It is notable that sowing in 2018 occurred approximately one month later in the year than did sowing in the respective years of 2016 and 2017. The task of choosing an optimal sowing time depended on a number of factors such as weather and climate conditions and forecasts, soil moisture and rainfall patterns. As such, domain experts responsible for sowing the trials use their best judgement, combined with the data at hand, to decide on sowing date each year.”
2. We have added extensive discussion around the effects of rainfall on soil, as well as soil characteristics and permeability, on pages 9, 12 and 16. This discussion cites relevant literature and we hope brings more light to the topics in question.
3. We have renamed the final section of the article from ‘Future Directions’ to ‘Limitations and Future Directions’. There are many limitations listed in this section, though it may not have been obvious from the title, apologies for that. Some of the limitations listed include:
- A wider range of phenotypic traits could be analysed, such as number of spikers or developmental stages could be studies.
- Segmentation accuracy can always improve.
- Senescence quantification has not been considered in our work but could be in the future.
- We were only able to monitor a relatively small amount of environmental variables which could be increased in the future.
- Our field trials were of a relatively small size, allowing only for a few replications of each sample.
Reviewer 2 Report
Comments and Suggestions for Authors
Based on the study investigating the growth and yield performance of wheat varieties in Australian winter wheat trials under varying conditions across different years, with particular attention to the influence of water availability, fertilization, and environmental factors on crop performance, I recommend the following revisions:
1. The manuscript would benefit from a comprehensive elucidation of the methodologies employed for data collection, processing, and quality assurance to ensure data reliability and consistency. Furthermore, a meticulous exposition of the statistical analysis methodologies is warranted to facilitate readers' comprehension of the data processing workflows and analytical outcomes.
2. Acknowledging the pivotal role of environmental factors in shaping crop growth and yield, a thoroughgoing investigation into crop performance across diverse environmental conditions is advisable. This may encompass discussions on crop growth traits and adaptive mechanisms under stressors such as drought or high temperatures, alongside strategies for adjusting agronomic practices to ameliorate the impact of adverse environmental factors.
3. With respect to the selection of crop varieties and breeding strategies, a deeper exploration of varietal performance differentials under varying environmental contexts is warranted. Additionally, an exploration of breeding methodologies aimed at enhancing crop adaptability and stress resilience is pertinent.
4. It is prudent to delineate clear future research trajectories and emphases in the concluding segment of the manuscript. This may encompass delineating avenues for further elucidating the interplay between crops and the environment, fostering the development of crop varieties tailored to adverse environmental exigencies, and refining agronomic management practices, among others.
Comments on the Quality of English LanguageMinor editing of English language required.
Author Response
It appears that all of Reviewer 2s comments are about the general quality of the manuscript in terms, mostly, of discussion and exploration. However there does not appear to be any recommendations for specific changes to the manuscript. Due to this we hope that the recommendations of the other two reviewers have helped us to improve all aspects considerably. Nevertheless, we will attempt to respond to each point individually. Below are the key concepts of each point from this reviewer
- Explanation of methodologies and description of statistical tools.
- Discussion on crop performance and growth traits across diverse environmental conditions.
- Deeper exploration of varietal performance differentials.
- Outline future work
1. The materials and methods section outlines all stages of data collection, processing and quality assurance. First the design of the field trials is discussed, then the image acquisition methods, including steps taken to ensure image quality and consistency. This trend continues through a number of subsections outlining each step of the methodology. Upon the advice of another reviewer we have added more information about choice of sowing dates, which we think further strengthens this section. For statistical tools, we have created an entire subsection called ‘Statistical Analysis’ where Equations 2-4 outline in detail the way that effect sizes are calculated and what they represent. Figures 15 and 16 are also purpose-built visualisations to demonstrate statistical differences over time.
2. Crop growth traits under diverse environmental conditions is discussed throughout the article.
3. While some discussion around varietal performance differentials has been provided, see Figure 11 and the respective discussion, it would be impossible to provide an exhaustive analysis of all varieties. That discussion would also deviate from the focus of the paper. That said, there is a supplementary file attached with the main article that shows performance differences for all varieties in terms of the growth indices measured. And while in-depth discussion for every variety pair is not included, all data is available, transparent and visualised neatly.
4. We have renamed the final section from ‘Future Directions’ to ‘Limitations and Future Directions’ and hope that the future research trajectories are clear in this section. Some of those listed include
- A wider range of phenotypic traits could be analysed, such as number of spikers or developmental stages could be studies.
- Segmentation accuracy can always improve.
- Senescence quantification has not been considered in our work but could be in the future.
- We were only able to monitor a relatively small amount of environmental variables which could be increased in the future.
- Our field trials were of a relatively small size, allowing only for a few replications of each sample.
Reviewer 3 Report
Comments and Suggestions for Authors
Nitrogen (N) is pivotal to crop yield, and the application of N fertilizer in crop production systems is a crucial aspect of modern crop management practices and one of the determining factors to increase crop yield and thereby keeping pace with human population increase. Excessive N fertilizer input is detrimental to environment and human health, causing eutrophication and contamination of water resources, ozone and air quality, greenhouse gas emission, soil acidification, and losses of biological diversity. This study on relationship between crop growth and response to N, and environmental conditions should be of interest to many readers of Sustainability.
In this paper, in-season, image-based, phenotyping studies of three wheat field trials in three years were designed and evaluated for canopy coverage, height and vigor versus time, variety and treatment were derived and statistically analyzed. As a result, an early application of nitrogen and timely and sustained rainfall proved to be the best combination and wheat yield appeared strongly positively correlated with canopy traits of coverage and vigor under well-watered conditions.
The study was considered to be a detailed investigation of multiple factors such as non-invasive acquisition and analysis of images, environmental data (rainfall, temperature and soil composition) was also collected in order to quantify the relationship between crop growth and response to nitrogen, and environmental conditions from a multifaceted perspective. In addition, the study was robust in that it examined multiple varieties over multiple years. The manuscript is well written and acceptable for publication after minor review.
Minor revision:
Line 19-30 references are needed
Line 40-72 The cited references should identify the author, such as ‘In [8], the authors investigated effect of’
Line 180-109 Why is the sowing date in 2018 is one month earlier than in previous years?
Fig. 4. The information of Fig should be integrity and simplicity. In Fig. 4 d, the author had distinguished the year by color, so the and it is not necessary to mark the year with arrows. In Fig. 4 a-c, the sowing day, first and last days of imaging were presented in Fig. 4 d and repeated again.
Line 190 Field conditions: author need add the discussion of the effects of rainfall and temperature to soil properties (i.e., silt, clay, NH4+, etc), instead of describe the results simply.
Line 258-265 add reference, Drying after anthesis will make the plant senescence.
Line 308 author need discuss the relationship between canopy and NUE, and the mechanism of increasing canopy to improve NUE.
Line375-376 Some conclusions and assumptions need to be discussed in depth in combination with references.
Some mistake in Figure: Author needs to check whether the expression in the Fig is standardized (such as ammonium nitrogen is NH4+-N instead of NH4)
Author Response
Thank you for your valuable and in-depth comments. We have responded to each point below.
1. Thank you for bringing this to our attention. These important lines in the introduction were definitely lacking in citations. A number of citations have now been added to provide evidence for each claim relating to the motivation of the paper.
2. This has been amended, thank you.
3. Thank you for highlighting this important difference in the sowing dates between trials. We have added the following sentences to the materials and methods section
“It is notable that sowing in 2018 occurred approximately one month later in the year than did sowing in the respective years of 2016 and 2017. The task of choosing an optimal sowing time depended on a number of factors such as weather and climate conditions and forecasts, soil moisture and rainfall patterns. As such, domain experts responsible for sowing the trials use their best judgement, combined with the data at hand, to decide on sowing date each year.”
4. Thank you for this suggestion. Upon reflection the figure is indeed over-crowded and there is redundancy in information. The arrows have been removed from the figure. We have however decided to keep the lines for sowing and imaging dates in both figure to allow ease of comparison between both figured for the reader. Removing these from Figure 4d may increase the cognitive burden of the viewer through having to memorise dates when changing focus from one subfigure to the next.
5. We have added extensive discussion around the effects of rainfall on soil, as well as soil characteristics and permeability, on pages 9, 12 and 16. This discussion cites relevant literature and we hope brings more light to the topics in question.
6. Thank you for this valuable suggestion. We have included a reference to Yang & Zhang 2005 in New Phytologist which provides references to nine articles demonstrating that post-anthesis water deficits result in early senescence.
7. We have added information about how growth indices such as canopy height and coverage are measures of clorophyll content and as such can be used to assess nitrogen status of the canopy, and included reference to the relevant literature
8. This question also relates to soil characteristics so the response to point 5 is relevant here also.
9. These subfigure headings have been amended, thank you.